# Brow and Masticatory Muscle Activity Senses Subjective Hedonic Experiences during Food Consumption

**DOI:** 10.3390/nu13124216

**Published:** 2021-11-24

**Authors:** Wataru Sato, Akira Ikegami, Sayaka Ishihara, Makoto Nakauma, Takahiro Funami, Sakiko Yoshikawa, Tohru Fushiki

**Affiliations:** 1Psychological Process Team, Guardian Robot Project, RIKEN, Kyoto 619-0288, Japan; 2Field Science Education and Research Center, Kyoto University, Kyoto 606-78502, Japan; yoshikawa.sakiko.4n@kyoto-u.ac.jp; 3San-Ei Gen F. F. I., Inc., Osaka 561-8588, Japan; akira-ikegami@saneigenffi.co.jp (A.I.); sayaka-ishihara@saneigenffi.co.jp (S.I.); m-nakauma@saneigenffi.co.jp (M.N.); tfunami@saneigenffi.co.jp (T.F.); 4Faculty of the Arts, Kyoto University of the Arts, Kyoto 606-8271, Japan; 5Faculty of Agriculture, Ryukoku University, Otsu 520-2194, Japan; tfushiki@agr.ryukoku.ac.jp

**Keywords:** facial electromyography (EMG), food, liking, mastication, wanting, valence

## Abstract

Sensing subjective hedonic or emotional experiences during eating using physiological activity is practically and theoretically important. A recent psychophysiological study has reported that facial electromyography (EMG) measured from the corrugator supercilii muscles was negatively associated with hedonic ratings, including liking, wanting, and valence, during the consumption of solid foods. However, the study protocol prevented participants from natural mastication (crushing of food between the teeth) during physiological data acquisition, which could hide associations between hedonic experiences and masticatory muscle activity during natural eating. We investigated this issue by assessing participants’ subjective ratings (liking, wanting, valence, and arousal) and recording physiological measures, including EMG of the corrugator supercilii, zygomatic major, masseter, and suprahyoid muscles while they consumed gel-type solid foods (water-based gellan gum jellies) of diverse flavors. Ratings of liking, wanting, and valence were negatively correlated with corrugator supercilii EMG and positively correlated with masseter and suprahyoid EMG. These findings imply that subjective hedonic experiences during food consumption can be sensed using EMG signals from the brow and masticatory muscles.

## 1. Introduction

Sensing subjective hedonic experiences (e.g., liking) during food consumption using physiological signals would offer practical and theoretical advantages. In terms of practicality, subjective hedonic responses to food are important for individuals, making everyday life more pleasurable [1,2] and reducing stress [3], and for companies aiming to develop new food products [4]. However, subjective ratings have some intrinsic limitations, including their inherently subjective nature and various biases [5]. Measuring physiological signals associated with hedonic experiences can complement subjective ratings by providing objective and unbiased data [6]. In terms of theoretical significance, understanding the physiological or behavioral correlates of subjective hedonic experiences during food consumption can provide unique clues regarding the evolutionary process and underlying neural mechanisms of human emotion [7].

Despite the importance of clarifying the physiological correlates of hedonic experiences during solid food consumption, empirical research on this issue remains scarce. To the best of our knowledge, only one previous study has investigated this issue by assessing participants’ ratings of liking, wanting, valence, and arousal while they consumed various gel-type solid foods [8]. The researchers recorded participants’ physiological activity, including facial EMG measured from the corrugator supercilii, zygomatic major, masseter, and suprahyoid muscles. Participants were instructed to first masticate for 5 s and then to taste without masticating for 5 s, and physiological signals during the tasting period were analyzed. The results showed that ratings of liking, wanting, and valence were negatively associated with corrugator supercilii EMG activity. This result was consistent with the findings of prior literature. For example, several previous studies that measured EMG from several facial muscles while participants ingested liquid stimuli (e.g., a sucrose solution) found a negative correlation between valence/liking ratings and corrugator supercilii EMG activity [9,10,11]. Several other studies also recorded EMG from facial muscles including the corrugator supercilii, while participants engaged in various non-eating tasks (e.g., observing images and films, listening to sounds, and reading words) and reported negative correlations between valence ratings and corrugator supercilii EMG activity [12,13,14,15,16,17]. It is thought that associations between subjective emotional experiences and facial muscle responses may reflect the read-out and feedback induction mechanisms that play a role in subjective emotional states [18]. These results imply that EMG activity recorded from the corrugator supercilii muscles, which correspond to brow lowering [18], is a physiological correlate of hedonic experience during the consumption of food.

However, the merits of the previous study reporting subjective–physiological concordance during solid food consumption [8] were limited in the task required that participants stop chewing while physiological data were being recorded. Although this approach may have reduced artifacts related to mastication, it may also have produced an unnatural manner of eating that could induce artificial corrugator supercilii EMG activity and suppress zygomatic major EMG activity. Several prior studies testing non-food stimuli have reported that valence ratings were positively associated with EMG activity recorded from the zygomatic major muscle [12,13,14,15,16,17]. Based on these data, we hypothesized that hedonic experiences could be associated not only negatively with corrugator supercilii EMG but also positively with zygomatic major EMG during food consumption with mastication.

Furthermore, the previous study’s interruption of mastication precluded any investigation into associations between hedonic experiences and EMG signals of the masticatory muscles, including the masseter and suprahyoid muscles [19,20]. A previous study has analyzed videotaped facial reactions in human infants and non-human infant/adult primates during the ingestion of liquids of various tastes, and found that tongue protrusions and gapes to sucrose and quinine, respectively, were elicited universally across species [21]. Studies consistently observed mouth motions in response to sucrose solution in human infants [22,23,24,25,26] and adults [27,28]. These data suggest that mouth movements related to consumption could be modulated by hedonic experiences during tasting. Still, these studies did not collect data regarding the participants’ subjective experiences. Based on these findings, we also hypothesized that subjective hedonic ratings could be positively associated with EMG activity recorded from the masticatory muscles during food consumption.

To test these hypotheses, we measured subjective hedonic ratings and physiological signals while participants consumed solid food with mastication. As stimuli, we used bite-sized gel-type foods (water-based gellan gum jellies) of diverse flavors. Participants provided subjective ratings of liking, wanting, valence, and arousal, and we measured EMG from the corrugator supercilii, zygomatic major, masseter, and suprahyoid muscles as in a previous study [8]. We also exploratorily recorded the skin conductance response (SCR), heart rate (HR), and nose-tip temperature, which mainly reflect autonomic nervous system activity and emotional arousal [17,29,30,31,32], although previous studies have not reported consistent responses to food or liquid stimuli for these signals [8,33,34,35]. We analyzed intra-individual correlations [36,37,38,39] between the participant’s subjective ratings and physiological activity, and tested the significance of correlations using second-level group analyses. We expected that liking, wanting, and valence ratings would be negatively associated with corrugator supercilii EMG and positively associated with zygomatic major, masseter, and suprahyoid EMG during the consumption of solid food. Body mass index (BMI) was also assessed and exploratory analyses of the relationship of BMI with subjective–physiological correlations were conducted, as previous studies have reported mixed findings regarding the relationship between hedonic responses to food stimuli and BMI [40,41]. In short, we aimed to clarify the association between subjective hedonic ratings and facial EMG signals during the natural consumption of solid foods.

## 2. Materials and Methods

### 2.1. Participants

Thirty young Japanese adults (16 females; mean ± *SD* age, 25.1 ± 6.5 years) participated in this study. We estimated the sample size using an *a priori* power analysis using G*Power 3.1.9.2 software [42]. We assumed a one-sample *t*-test (two-tailed) with an *α* level of 0.05, power of 0.90, and effect size *d* of 0.7. The results showed that 24 participants would be needed. Participants aged between 20–40 years were recruited via online advertisements; we excluded older participants because previous psychophysiological studies have reported differences in the degree of subjective–physiological concordance in emotional responses between young and older age groups [43,44]. All participants fasted for >3 h before the experiments and were not obese (BMI, <30, mean ± *SD*, 22.1 ± 2.8 kg/m^2^). All participants gave written informed consent. Ethical approval for this study was obtained from the Ethics Committee of the Unit for Advanced Studies of the Human Mind, Kyoto University (30-P-6) on 21 September 2018. All experiments were conducted following institutional ethics provisions and the Declaration of Helsinki.

### 2.2. Stimuli

In this study, we prepared simple water-based gels, with sweet and sour tastes and various flavors, using low-acylated gellan gum. Hydrocolloid gels can provide varying textures and are therefore a useful model of solid food [19,45,46]. Low-acylated gellan gum is one of these hydrocolloids, which is generally used as gelling agent in the manufacture of gel-type foods, such as water-based dessert jellies, fillings, and puddings [47]. Gellan gum forms gels with hard and brittle texture similar to agar gels, and with better flavor release than agar gels [48].

We prepared bite-sized gel-type solid food materials of nine flavors and presented these twice to each participant (Table 1; Figure 1). For practice, we prepared another three food items. All stimuli contained 10% sucrose, 0.2% anhydrous citric acid, 0.03% trisodium citrate, 0.1% calcium lactate, and 0.35% or low-acetylated gellan gum (Kelcogel; San-Ei Gen F.F.I., Osaka, Japan). Each stimulus contained a specific flavoring agent (Table 1). These flavor compounds were provided by San-Ei Gen F.F.I., Osaka, Japan. All of the flavor compounds are used in commercial foods. The flavoring agent concentrations were established in a series of preliminary experiments with participants who did not take part in the final experiment. Because food preferences generally differ between individuals [49], we were interested in testing individual specific subjective–physiological concordance. The number of food stimuli (i.e., 3 + 18) was determined through preliminary experiments so that participants with a lighter appetite would be able to comfortably consume all of the jellies.

To prepare the gel stimuli, a mixture of sucrose and low-acetylated gellan gum (Kelcogel; San-Ei Gen F.F.I., Osaka, Japan) was added to deionized water at 90 °C in 500-mL glass beakers, and was stirred at 1300 rpm for 10 min at the same temperature. Next, the calcium lactate, anhydrous citric acid, and trisodium citrate were added to the solutions. The solutions were infused into plastic cups (65 mm in diameter, 25 mm in height) in which cylindrical glass molds (20 mm in diameter, 10 mm in height) were set. The cups containing the solutions were sealed hermetically, heated at 85 °C for 30 min, and refrigerated at 8 °C for 1 h. The prepared gels were 20 mm in diameter and 10 mm in height. The fracture force and fracture strain of these gels were determined by compressing the gels on a metal stage using a 75-mm diameter aluminum plate, operating at a crosshead speed of 10 mm/s at 20 °C, and were measured by using a TA XT-2i texture analyzer (Stable Micro Systems, Godalming, UK). The results showed the fracture force of 17.0 ± 0.9 N, and the fracture strain of 45.3 ± 2.2%.

### 2.3. Procedure

Participants were tested individually in an electrically shielded soundproof chamber (Science Cabin; Takahashi Kensetsu, Tokyo, Japan). The room temperature was maintained at 23.5–24.5 °C over the course of the experiment and was monitored by using a TR-76Ui (T&D, Matsumoto, Japan). Experiments were controlled by Presentation 14.9 software (Neurobehavioral Systems, Berkeley, CA, USA) on a Windows computer (HP Z200 SFF, Hewlett-Packard Japan, Tokyo, Japan). Visual stimuli were displayed on a 19-inch computer screen (HM903D-A; Iiyama, Tokyo, Japan).

At the beginning of the experiment, participants were informed that subjective hedonic ratings and electric physiological signals would be measured while eating. A dish holding the nine food stimuli was placed on the table in front of a monitor. This dish was exchanged for another dish with nine different food stimuli during a recess. The stimuli were placed in a row on 8-cm plastic disposable spoons so that participants could eat them with one hand in a predefined order. All stimuli were prepared about 10 min prior to the experiment. After three practice trials, 18 experimental trials were performed, with a short recess after half of the trials. The order of stimulus presentation was pseudo-randomly determined with no repetition of general hedonic quality. The inter-trial interval randomly varied between 20 and 30 s.

Each trial began with a small white cross presented for 3 s against a black background on the screen, as a warning cue. Next, a large red cross was presented for 10 s to signal the consumption period. Finally, a rating display with four scales (liking, wanting, valence, and arousal) was presented until the ratings were completed. The participants were instructed to (1) prepare to consume the stimuli by holding the spoon close to their mouth when the white cross appeared; (2) consume each stimulus as soon as the red cross appeared and keep chewing without swallowing while the red cross was displayed; and (3) swallow the stimulus and rate their hedonic experiences during consumption by pressing keys when the response panel was shown. Four 9-point rating scales were displayed simultaneously in the fixed order of liking, wanting, valence, and arousal. Liking and wanting ratings were made using these terms as labels and lines with numbers and wording indicating 1 (dislike) to 9 (like) and 1 (do not want to eat) to 9 (want to eat), respectively. For valence and arousal ratings, these terms as labels and numbers with images of self-assessment manikins [50] were displayed. After the ratings were completed, participants were instructed to rinse their mouth with mineral water.

### 2.4. Physiological Data Recording

Facial EMG data were recorded using Ag/AgCl electrodes 0.7 cm in diameter (Prokidai, Sagara, Japan) with a 1.5-cm inter-electrode spacing, an EMG-025 amplifier (Harada Electronic Industry, Sapporo, Japan), a PowerLab 16/35 data acquisition system, and LabChart Pro v8.0 software (ADInstruments, Dunedin, New Zealand). The data were filtered online with a band-pass of 20–400 Hz and digitized at a sampling rate of 1000 Hz. The electrodes were placed on the corrugator supercilii, zygomatic major, masseter, and suprahyoid muscles according to guidelines [51,52] and previous studies [19,20] (Figure 2). A ground electrode was placed on the forehead.

SCR data were recorded using Ag/AgCl electrodes 1.0 cm in diameter (Vitrode F; Nihonkoden, Tokyo, Japan), a Model 2701 BioDerm Skin Conductance Meter (UFI, Morro Bay, CA, USA), and the same data acquisition system and software as with the EMG recording. The data were sampled at 1000 Hz. The electrodes were attached to the palmar surface of the medial phalanges of the left index and middle fingers, according to guidelines [29,53].

HR was recorded using the same apparatus as for the SCR recording. Electrodes were positioned on the left wrist and left ankle according to the guideline for electrocardiography [54]. The software automatically calculated beats per minute, which were sampled at 1000 Hz.

Nose-tip temperature data were recorded using an infrared thermal imaging camera FLIR A655sc and Research IR Max v4.40 software (FLIR Systems, Wilsonville, OR, USA). The camera was placed next to the computer screen and set to capture the entire face of each participant. The data were sampled at 50 Hz with a spatial resolution of 640 horizontal × 480 vertical pixels.

For artifact removal, mastication-related jaw movements (*x*-, *y*-, *z*-axial acceleration) were measured using an accelerometer (Bio Research Center, Nagoya, Japan) attached at the jaw mentum. Video were recorded unobtrusively using a digital web camera (HD1080P; Logicool, Tokyo, Japan).

### 2.5. Data Analysis

#### 2.5.1. Preprocessing

EMG data were analyzed using Psychophysiological Analysis Software 3.3 (Computational Neuroscience Laboratory, Salk Institute) and in-house programs on MATLAB 2020a (MathWorks, Natick, MA, USA). The data were sampled during the baseline period for 0.5 s immediately before the consumption period and during the consumption period for 10 s for each trial. The data for each trial were rectified, baseline-corrected to the average value over the baseline period, and averaged over the consumption period. The values for each stimulus were then standardized to *z* scores for each individual. For SCR, the maximum value during 1.5–10 s of the consumption period was calculated for each trial and then standardized for each individual. For HR, the baseline-corrected, averaged, and standardized values were calculated for each trial, but the data were not rectified. For the nose-tip temperature data, thermal images were first analyzed using Research IR Max v4.40 software (FLIR Systems, Wilsonville, OR, USA). As in a previous study [17], the data were extracted from a 3 × 3-pixel region of interest (ROI) created on the nose tip. The data were then baseline-corrected, averaged, and standardized in the same way as for HR analysis.

#### 2.5.2. Statistical Analysis

To test the individual-level concordance between subjective hedonic experiences and physiological activity, Pearson’s product-moment correlation coefficients (*r*-values) were calculated between subjective ratings and physiological signals for each participant. We made *a priori* predictions and analyzed the relationships between the ratings of liking/wanting/valence and EMG measured from the corrugator supercilii/zygomatic major/masseter/suprahyoid muscles. These analyses were planned and conducted independently, so there was no adjustment for multiple testing [55,56,57]. We also tested other relationships in the same manner for descriptive purposes. The *r*-values were normalized using Fisher transformation and analyzed using one-sample *t*-tests (two-tailed) to test for significant mean differences from 0, as in a previous study [8]. Such two-stage random-effects analyses can show the generalizability across individuals [58]. One participant frequently changed head position, making it difficult to analyze the thermal images, so the nose-tip temperature data of this participant were removed from the analyses. In addition, Pearson’s correlation coefficients were exploratorily calculated between the *r*-values and BMI. To visually illustrate the relationships between subjective hedonic ratings and physiological activity at the group level, we depicted the group-mean values and regression lines for subjective ratings and physiological signals. To test the possibility that mastication could influence EMG data from the corrugator supercilii or zygomatic major muscle, one-sample *t*-tests after Fisher transformation were conducted for these EMG data after additional preprocessing to adjust for (regress out) the effects of EMG activity recorded from the masseter and suprahyoid muscles and the *x*-, *y*-, *z*-axial accelerometer data recorded at the jaw mentum. Results were considered significant at *p* < 0.05.

## 3. Results

Appendix A list the mean ± *SD* subjective hedonic ratings and physiological activity in response to each stimulus. The correlation coefficients (*r*-values) between the ratings and physiological signals during food consumption were calculated for each participant to analyze intra-individual subjective–physiological concordance during food consumption (Figure 3). Next, the *r*-values after the Fisher *z* transformation were subjected to one-sample *t*-tests against a value of zero. The results revealed that the subjective ratings of liking, wanting, and valence were significantly negatively associated with corrugator supercilii EMG activity (*t* > 4.04, *p* < 0.001, *d* > 0.73) and positively associated with masseter and suprahyoid EMG activity (*t* > 2.31, *p* < 0.05, *d* > 0.37) (Table 2). Figure 4 presents these relationships as group-mean values. No other significant association between subjective ratings and physiological activity was found (*p* > 0.05).

Exploratory analysis of the correlation between BMI and each *r*-value of intra-individual subjective–physiological concordance was conducted. No significant correlation was found (|*r*| < 0.25, *p* > 0.10).

To test the possibility that mastication could affect the association between subjective ratings and facial EMG recorded from the corrugator supercilii or zygomatic major muscles, additional preprocessing adjusting for (regressing out) the effects of EMG activity recorded from the masseter and suprahyoid muscles and the *x*-, *y*-, *z*-axial accelerometer data recorded at the jaw mentum were conducted on these EMG data. The results of one-sample *t*-tests after the Fisher *z* transformation for these adjusted EMG data revealed the same pattern of significant associations between subjective ratings and non-adjusted EMG signals. Specifically, the subjective ratings of liking, wanting, and valence were significantly and negatively associated with corrugator supercilii EMG activity (*t* > 4.43, *p* < 0.001, *d* > 0.80). None of these ratings was significantly associated with zygomatic major EMG activity (*t* < 0.46, *p* > 0.655, *d* < 0.08).

## 4. Discussion

In this study, we investigated associations between subjective hedonic experiences and facial EMG activity during the natural consumption of solid foods. Participants consumed gel-type food stimuli of diverse flavors with mastication. Participants rated the stimuli in terms of liking, wanting, valence, and arousal, and EMG signals were recorded from the corrugator supercilii, zygomatic major, masseter, and suprahyoid muscles. Some autonomic nervous system parameters, including SCR, HR, and nose-tip temperature, as well as BMI, were also explored.

Our results for the intra-individual correlations between subjective ratings and physiological signals revealed that the ratings of liking, wanting, and valence were negatively associated with corrugator supercilii EMG activity during the consumption of gel-type foods. The same results were found for the data adjusted for masticatory EMG and acceleration. These results are consistent with the findings of a prior study [8]. The researchers in that study asked participants to taste the food without mastication, so the current study expands upon this prior study by demonstrating that relaxation of corrugator supercilii EMG can reflect hedonic experiences during natural eating. Our results also revealed that the ratings of liking, wanting, and valence were not significantly associated with zygomatic major EMG activity with and without adjustment for masticatory EMG and acceleration, which is consistent with previous findings [8]. Although the results of several previous studies testing non-eating tasks revealed that subjective valence ratings were associated with EMG activity measured from both the corrugator supercilii and zygomatic major muscles (e.g., [12]), some studies found that subjective ratings were associated with EMG for only one of these muscles [14,59]. Taken together, our results imply that subjective hedonic experiences during eating are more evidently associated with EMG signals measured from corrugator supercilii EMG compared with zygomatic major EMG.

Furthermore, our results revealed that the ratings of liking, wanting, and valence were also positively associated with EMG signals recorded from the masseter and suprahyoid muscles during the consumption of food materials with mastication. These findings expand on those from the previous study, which recorded EMG from these muscles during eating but prohibited mastication during physiological data acquisition [8]. To the best of our knowledge, the present study provides the first evidence that hedonic experiences during food consumption are associated with masticatory EMG activity. The results may be in line with prior findings that the ingestion of liquid stimuli of various tastes elicited mouth movements in humans and animals (e.g., [21]). Several studies also reported that mastication could be related to a wide range of eating behaviors, including appetite and food intake (e.g., reduced hunger ratings and energy intake with prolonged mastication) [60]. A recent study reported that stronger masticatory performance was associated with greater retronasal aroma release [61], which may be the mechanism underlying stronger mastication for hedonically preferred foods. Although EMG signals recorded from the masticatory muscles were rarely mentioned in studies searching for physiological correlates of subjective emotional experiences [30,62], and hedonic experiences received little attention as a modulatory factor in mastication [63], these data collectively imply that subjective hedonic experiences during food consumption may be associated with masticatory muscle EMG.

Our results from autonomic nervous system measures, including SCR, HR, and nose-tip temperature, revealed no significant associations with hedonic ratings. These results are consistent with previous findings [8,34] and suggest that autonomic nervous activity may not be reliably correlated with subjective hedonic experiences during food consumption. One possible explanation for this is that the food stimuli used in this study had a weaker ability to reveal associations between subjective arousal and autonomic nervous system activity. Further studies using different types of food stimuli are needed to clarify this issue.

Our exploratory analyses revealed that subjective–physiological concordance was not significantly correlated with BMI. This is in line with a previous meta-analysis, which found that subjective hedonic responses to food stimuli were not consistently associated with BMI [40]. The lack of association between subjective–physiological and obesity in our study may be attributable to an insufficient sample size; further investigations are needed.

Our results demonstrating facial EMG correlates of hedonic experiences during food consumption have practical implications. First, because hedonic experiences during eating induce pleasure [1,2] and reduce stress [3], it may be useful to quantify individuals’ hedonic experiences using facial EMG signals. Specifically, such physiological recording would be useful for participants who are unable to report their hedonic experiences, such as infants and individuals with dementia. Second, objectively sensing hedonic experiences during eating could be useful for food companies aiming to develop new food products. Although these companies mainly rely on subjective ratings, it has been shown that participants’ subjective responses can be biased in experimental situations toward confirming researchers’ hypotheses [64,65]. Facial EMG recordings may complement subjective ratings and provide unbiased measurements of hedonic reactions to food products.

Our results also have theoretical implications. First, our finding that facial EMG correlates with subjective hedonic experiences during eating in humans suggest that such subjective–physiological concordance may exist across species. Although previous comparative studies on this issue mainly utilized liquid stimuli and video recordings of facial reactions [21], EMG recordings from the brow and masticatory muscles could be collected during the consumption of solid foods, and may provide new insight into the hedonic experiences of other animals. Second, the current data showing subjective–physiological concordance suggest that brow and masticatory muscle activity may reflect the feedback induction mechanism for subjective hedonic experiences [18] during eating. The data suggests that the manipulation of brow and masticatory muscles (i.e., relaxing the brow and strengthening chewing) may modulate hedonic experiences during eating. In the literature examining emotional responses to non-food stimuli, the manipulation of facial muscles was shown to modulate emotional experiences [66] and to influence depression [67]. It may be useful to explore the causal relationship between the physiological and subjective components of eating, given the correlations found in the present study.

### Limitations

Several limitations of this study should be acknowledged. First, we did not record physiological data during the swallowing process, which could also be associated with subjective hedonic experiences. A recent study has reported that surface electrodes placed on the neck region detected swallowing activity [68]. Future studies involving placement of electrodes in this region and measurement of EMG data at the late stage of food consumption may broaden the physiological correlates of subjective hedonic experiences during the consumption of food.

Second, as we used only gel-type food materials, the generalizability of our findings to other solid foods is unclear. For example, although the gel-type foods used herein all had the same texture, previous studies have shown that taste perception can be modified by the textural properties of foods [69]. Also, our stimuli were all colorless; several previous studies demonstrated that food color affects the processing of taste and flavor, as well as hedonic value [70]. The systematic manipulation of these parameters in gel-type food materials, and presentation of other types of solid foods, are warranted to further investigate subjective–physiological concordance during eating.

Finally, our participants were all normal young adults; hence, the generalizability to other populations is unclear. In addition, we only tested the modulatory effect of BMI. Previous studies have suggested the importance of other variables. For example, one study reported different subjective and facial EMG reactions to food images between clinical samples and healthy controls; for example, individuals with anorexia nervosa showed stronger corrugator supercilii EMG responses to food images than controls [71]. Some studies reported greater subjective–physiological concordance in emotional responses among older than young participants viewing emotional films [43,44]. Several studies found that both an individuals’ hunger [72] and food neophobia levels [73] modulated subjective hedonic reactions during the consumption of food. These data suggest that subjective–physiological concordance during food consumption may differ between samples and depend on the characteristics thereof; this should be investigated in future research.

## 5. Conclusions

Our results demonstrated that subjective ratings of liking, wanting, and valence were negatively associated with corrugator supercilii EMG and positively associated with masseter and suprahyoid EMG during food consumption with mastication. These findings imply that subjective hedonic experience during the consumption of food can be sensed using EMG signals from the brow and masticatory muscles.

## Figures and Tables

**Figure 1 nutrients-13-04216-f001:**
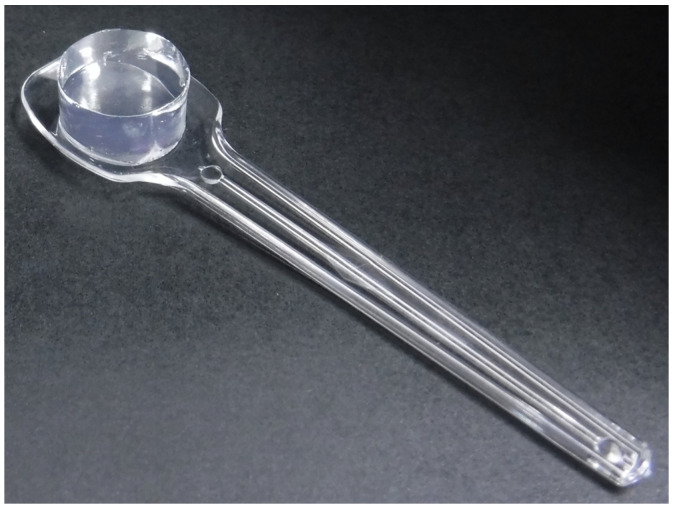
An example of the gel-type solid food stimulus.

**Figure 2 nutrients-13-04216-f002:**
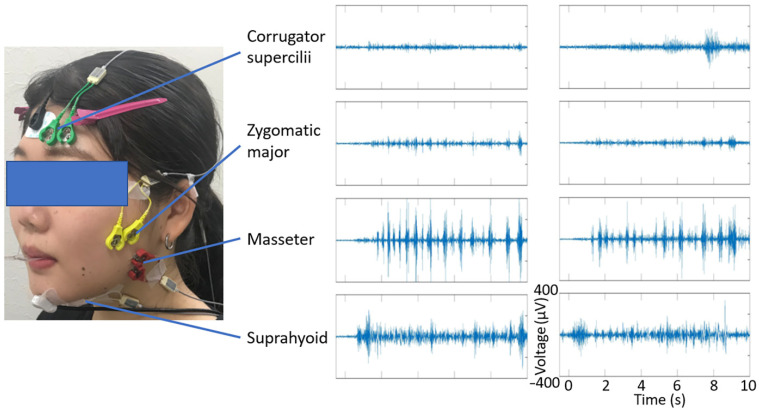
Illustrations of electrode placement (left) and representative data (right) for electromyography recording of the corrugator supercilii, zygomatic major, masseter, and suprahyoid muscles. The person shown is an amateur model who did not participate in the experiment and provided written consent to show her face in scientific journals.

**Figure 3 nutrients-13-04216-f003:**
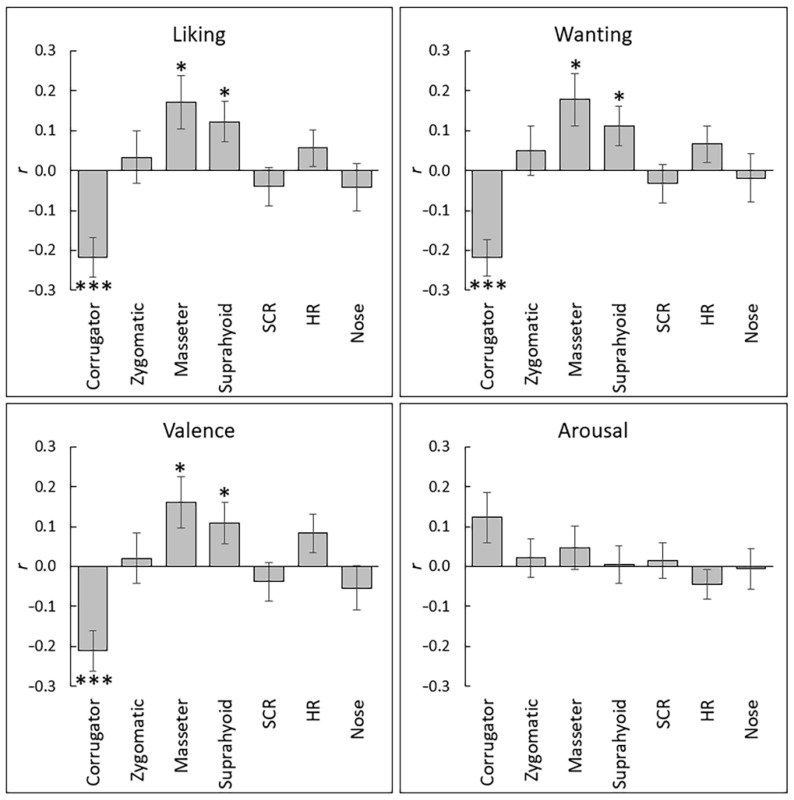
Mean (with *SE*) intra-individual correlation coefficients between subjective ratings and physiological responses across stimuli. Corrugator = corrugator supercilii; Zygomatic = zygomatic major; SCR = skin conductance response; HR = heart rate; Nose = nose-tip temperature. ***, *p* < 0.001; *, *p* < 0.05.

**Figure 4 nutrients-13-04216-f004:**
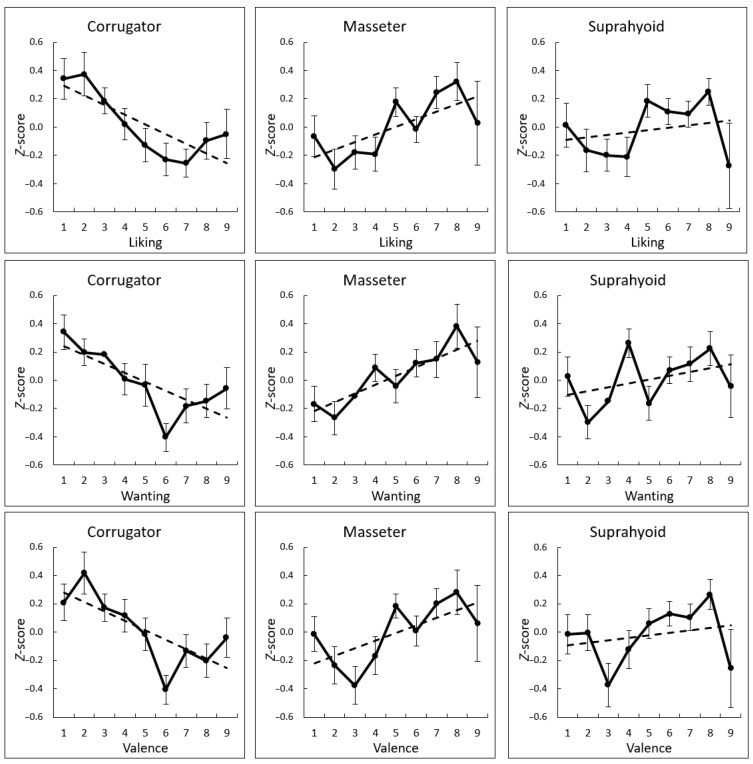
Group-mean (with *SE*) values and regression lines of subjective ratings (liking, wanting, and valence) and electromyography activity recorded from the corrugator supercilii, masseter, and suprahyoid muscles (standardized for each individual).

**Table 1 nutrients-13-04216-t001:** Characteristics of the gel-type solid food stimuli.

Hedonic Quality	Flavor Compound	Concentration (*w*/*w*%)	Odor
Negative	Isovaleric acid	0.0010	Sweaty
	(E)-2-nonenal	0.0005	Cucumber
	Indole	0.0010	Fecal
Neutral	Phenethyl alcohol	0.0100	Bread
	Acetoin	0.0100	Yogurt
	2,5-dimethylpyrazine	0.0100	Roast
Positive	Vanillin	0.0200	Vanilla
	Maltol	0.0200	Caramel
	Ethyl butyrate	0.0200	Pineapple

**Table 2 nutrients-13-04216-t002:** Results of one-sample *t*-test (two tailed) for intra–individual correlation coefficients between subjective ratings and physiological responses.

Subjective	Statistic	Physiological						
		Corrugator	Zygomatic	Masseter	Suprahyoid	SCR	HR	Nose
Liking	*t*	**4.35**	0.47	**2.38**	**2.35**	0.89	1.24	1.01
	*p*	**<0.001**	0.642	**0.024**	**0.026**	0.38	0.224	0.885
	*d*	**0.80**	0.09	**0.43**	**0.43**	0.16	0.23	0.18
Wanting	*t*	**4.47**	0.78	**2.62**	**2.19**	0.69	1.43	0.97
	*p*	**<0.001**	0.441	**0.014**	**0.037**	0.497	0.163	0.724
	*d*	**0.82**	0.14	**0.48**	**0.40**	0.13	0.26	0.18
Valence	*t*	**4.05**	0.35	**2.32**	**2.06**	0.80	1.76	0.42
	*p*	**<0.001**	0.732	**0.028**	**0.048**	0.432	0.09	0.68
	*d*	**0.74**	0.06	**0.42**	**0.38**	0.15	0.32	0.08
Arousal	*t*	2.04	0.42	0.73	0.16	0.34	1.27	0.72
	*p*	0.051	0.68	0.469	0.878	0.736	0.215	0.476
	*d*	0.37	0.08	0.13	0.03	0.06	0.23	0.13

Degrees of freedom are 29 for all except Nose (28). Significant results (*p* < 0.05) are in bold. Corrugator = corrugator supercilii; Zygomatic = zygomatic major; SCR = skin conductance response; HR = heart rate; Nose = nose-tip temperature.

## Data Availability

The data that support the findings of this study are available on request from the corresponding author.

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
