# Peer review of "Brow and Masticatory Muscle Activity Senses Subjective Hedonic Experiences during Food Consumption"

_nutrients, 2021, doi:10.3390/nu13124216_

Round 1

Reviewer 1 Report

Thank you for giving me the opportunity to read this paper that investigates the correlation between the subjective enjoyment of food and facial muscle reactions. The paper is well written and has a solid methodology. However, the central theme and implication of the paper could be better carved out by being more specific in the introduction and discussion. Further, I would like more information about sample characteristics.

Please find more concrete suggestions below.

Abstract

  • I would suggest to explain the term “masticating” and to give an example for “gel-type solid food”, since this is otherwise difficult to imagine.

Introduction

  • Line 37-38: “well-being” and “health management” is very broad, be more specific. This could underline the relevance of understanding emotional experience connected to food intake, e.g. in the context of obesity, emotional eating, and eating disorders. See for example Schnepper et al. 2020 (Fight, Flight, – Or Grab a Bite! Trait Emotional and Restrained Eating Style Predicts Food Cue Responding Under Negative Emotions) and Schnepper et al. 2021 (Bad mood food? Increased versus decreased food cue reactivity in anorexia nervosa and bulimia nervosa during negative emotions)
  • Line 56: “literature” has no plural
  • I would go a bit more into detail about the connection between emotional experience and facial muscle reactions, e.g. as described by “Effects of positive and negative affect on electromyographic activity over zygomaticus major and corrugator supercilii. Larsen JT, Norris CJ, Cacioppo JT. Psychophysiology. 2003 Sep; 40(5):776-85”
  • Line 73-75 & line 86-88: I would suggest to formulate directional hypotheses, since you have clear expectations on the direction of your effect, which you describe later

Method:

  • What were your inclusion and exclusion criteria for participation?
  • L 116: is “mine” meant to say “nine”?
  • Section 2.2.: I find it hard to imagine what the Stimuli actually tasted like, was the idea to imitate specific foods (e.g. chocolate, nuts, …)
  • The sample should be described in more detail. Did you measure any descriptives that could influence the strength of the relation? E.g. hunger, BMI, mood, …
  • Section 2.3. and 2.4. are very thoroughly written, all information needed is given
  • Section 2.5.2.: Did you correct for multiple testing?

Discussion:

  • Start the discussion with a short description about the aim of the study
  • Try to not repeat the findings of previous studies in the first paragraph, but instead interpret your own findings in the light of previous studies
  • Lines 320 – 331: Don’t interpret non-significant results
  • Lines 332 – 347: Again, be more specific about implications instead of using very broad terms like “happiness” and “good health”

Author Response

Thank you for your helpful suggestions on improving our manuscript. We have modified the manuscript accordingly. We have also made changes to the manuscript to reduce the similarity to previously published articles, as requested by the Editorial Office. In addition, a professional English-language editing service has made language-related changes to the manuscript (http://www.textcheck.com/certificate/p8DG4N). All revisions to the manuscript are marked using the "Track Changes" function.

Point 1
However, the central theme and implication of the paper could be better carved out by being more specific in the introduction and discussion.
Response
Based on your suggestion, we have modified the Interlocution and in the Discussion sections.

Point 2
Further, I would like more information about sample characteristics.
Response
As suggested, we have added additional information regarding the sample characteristics to the Materials and Methods section (p. 3).

Point 3
Abstract
I would suggest to explain the term “masticating” and to give an example for “gel-type solid food”, since this is otherwise difficult to imagine.
Response
In accordance with your suggestion, we have explained “masticating” and added an example of gel-type solid food in the Abstract (p. 1).

Point 4
Introduction
Line 37-38: “well-being” and “health management” is very broad, be more specific. This could underline the relevance of understanding emotional experience connected to food intake, e.g. in the context of obesity, emotional eating, and eating disorders. See for example Schnepper et al. 2020 (Fight, Flight, – Or Grab a Bite! Trait Emotional and Restrained Eating Style Predicts Food Cue Responding Under Negative Emotions) and Schnepper et al. 2021 (Bad mood food? Increased versus decreased food cue reactivity in anorexia nervosa and bulimia nervosa during negative emotions)
Response
As suggested, we have added an explanation of the significance of subjective hedonic responses to food to the Introduction (p. 1).

Point 5
Line 56: “literature” has no plural
Response
As suggested, we have made the correction (p. 2).

Point 6
I would go a bit more into detail about the connection between emotional experience and facial muscle reactions, e.g. as described by “Effects of positive and negative affect on electromyographic activity over zygomaticus major and corrugator supercilii. Larsen JT, Norris CJ, Cacioppo JT. Psychophysiology. 2003 Sep; 40(5):776-85”
Response
As suggested, we have added a discussion of the relationship between emotional experience and facial muscle reactions to the Introduction (p. 2). We have explained that the relationship has been observed in various non-eating tasks and was interpreted as read-out and feedback systems for subjective emotional states.

Point 7
Line 73-75 & line 86-88: I would suggest to formulate directional hypotheses, since you have clear expectations on the direction of your effect, which you describe later
Response
As suggested, we have added a directional hypotheses to the Introduction section (p. 2).

Point 8 
Method:
What were your inclusion and exclusion criteria for participation?
Response
We only recruited young adult participants (aged: 20–40 years) in this study. This information and study rationale was described in the Materials and Methods section (p. 3). We have added this as a study limitation to the Discussion section (p. 13). 

Point 9
L 116: is “mine” meant to say “nine”?
Response
As suggested, we have made the correction (p. 3).

Point 10
Section 2.2.: I find it hard to imagine what the Stimuli actually tasted like, was the idea to imitate specific foods (e.g. chocolate, nuts, …)
Response
As suggested, we have added more information regarding the gel-type food stimuli to the Materials and Methods section (p. 3). In addition, we have described the smell of the flavor agents in Table 1 (p. 4).

Point 11
The sample should be described in more detail. Did you measure any descriptives that could influence the strength of the relation? E.g. hunger, BMI, mood, …
Response
As suggested, we have added more details regarding the study participants to the Materials and Methods section (p. 3). We recruited young Japanese adults via online advertisements. We analyzed the modulatory effects of BMI on subjective–physiological associations as described in the Results section (p. 10) and Discussion (p. 12) sections. We did not assess other individual differences, such as hunger states and clinical symptoms, which has been added as a limitation to the Discussion section (p. 13). 

Point 12
Section 2.5.2.: Did you correct for multiple testing?
Response
We conducted a priori analyses regarding relationships between subjective ratings and facial EMG, without adjustment for multiple testing. Other relationships were also tested in a similar manner only for descriptive purposes. We have described these methods in the Materials and Methods section (p. 7).

Point 13
Discussion:
Start the discussion with a short description about the aim of the study
Response
As suggested, we have added a brief description of the study aims and overview to the Discussion section (p. 11).

Point 14
Try to not repeat the findings of previous studies in the first paragraph, but instead interpret your own findings in the light of previous studies
Response
As suggested, we have removed the details of previous findings from the said paragraph (p. 11).

Point 15
Lines 320 – 331: Don’t interpret non-significant results
Response
As suggested, we have removed our previous discussion of non-significant results from the Discussion section (p. 11).

Point 16
Lines 332 – 347: Again, be more specific about implications instead of using very broad terms like “happiness” and “good health”
Response
As suggested, we have added a discussion of the implications of the present findings to the Discussion section (p. 12).

We hope that this revised manuscript is deemed acceptable for publication in Nutrients. Thank you for your time and interest in our work.

Yours sincerely,

Wataru Sato

Reviewer 2 Report

Authors, Please note and address the following comments:

Introduction

The introduction is well written, but in my opinion, the purpose of this manuscript should be written in the last sentence.

Material and methods

How were study participants recruited? Were the evaluators neutral persons, neophiles, or neophobes? It is important and could have had an impact on the results.

I don’t understand why the authors chose this complex product for analysis. I don’t understand at all from this description that 18 samples were taken for the experiment. In my opinion, the authors should add an experiment plan to better understand the study. There were 18 samples for evaluation, which seems to be a lot for one analysis.

Conclusions

What are the practical and theoretical implications of the research?

The current conclusions are quite enigmatic.

Limitations

Research limitations should be a separate chapter.

References

References are cited according to journal rules.

Reviewer

Author Response

Thank you for your helpful suggestions on improving our manuscript. We have modified the manuscript accordingly. We have also made changes to the manuscript to reduce the similarity to previously published articles, as requested by the Editorial Office. In addition, a professional English-language editing service has made language-related changes to the manuscript (http://www.textcheck.com/certificate/p8DG4N). All revisions to the manuscript are marked using the "Track Changes" function.

Point 1
Introduction
The introduction is well written, but in my opinion, the purpose of this manuscript should be written in the last sentence.
Response
Based on your suggestion, we have added a description of the study purpose to the last sentence of the Introduction section (p. 3).

Point 2
Material and methods
How were study participants recruited? Were the evaluators neutral persons, neophiles, or neophobes? It is important and could have had an impact on the results.
Response
We recruited young adult participants aged 20–40 years via online advertisements. We have added this information and the study rationale to the Materials and Methods section (p. 3). We did not assess the personality of participants, including food neophobia; therefore, we have added this as a limitation to the Discussion section (p. 13).

Point 3
I don’t understand why the authors chose this complex product for analysis. I don’t understand at all from this description that 18 samples were taken for the experiment. In my opinion, the authors should add an experiment plan to better understand the study. There were 18 samples for evaluation, which seems to be a lot for one analysis.
Response
As suggested, we have described our rationale for the use of gel-type food products in the Materials and Methods section (p. 3). We used gel products because they are useful models of solid foods and popular.
The number of stimuli was determined based on the results of preliminary experiments. We have added this information to the Materials and Methods section (p. 3).
We agree that the stimuli were complex and further research is required. We have added this as a study limitation to the Discussion section (p. 13).

Point 4
Conclusions
What are the practical and theoretical implications of the research?
The current conclusions are quite enigmatic.
Response
As suggested, we have added the practical and theoretical implications of the present findings to the Discussion section (p. 12).

Point 5
Limitations
Research limitations should be a separate chapter.
Response
As suggested, we have added a separate chapter for study limitations to the Discussion section (p. 12).

Point 7
References
References are cited according to journal rules.
Response
As suggested, we have modified the References according to the journal guidelines.

We hope that this revised manuscript is deemed acceptable for publication in Nutrients. Thank you for your time and interest in our work.

Yours sincerely,

Wataru Sato